EMBO
Molecular Medicine

# Cyclic-di-GMP signalling and biofilm-related properties of the Shiga toxin-producing 2011 German outbreak *Escherichia coli* O104:H4

Anja M Richter[1],[†], Tatyana L Povolotsky[1],[†], Lothar H Wieler[2] & Regine Hengge[1],[*]

## Abstract

In 2011, nearly 4,000 people in Germany were infected by Shiga toxin (Stx)-producing *Escherichia coli* O104:H4 with > 22% of patients developing haemolytic uraemic syndrome (HUS). Genome sequencing showed the outbreak strain to be related to enteroaggregative *E. coli* (EAEC), suggesting its high virulence results from EAEC-typical strong adherence and biofilm formation combined to Stx production. Here, we report that the outbreak strain contains a novel diguanylate cyclase (DgcX)—producing the biofilm-promoting second messenger c-di-GMP—that shows higher expression than any other known *E. coli* diguanylate cyclase. Unlike closely related *E. coli*, the outbreak strain expresses the c-di-GMP-controlled biofilm regulator CsgD and amyloid curli fibres at 37°C, but is cellulose-negative. Moreover, it constantly generates derivatives with further increased and deregulated production of CsgD and curli. Since curli fibres are strongly proinflammatory, with cellulose counteracting this effect, high c-di-GMP and curli production by the outbreak O104:H4 strain may enhance not only adherence but may also contribute to inflammation, thereby facilitating entry of Stx into the bloodstream and to the kidneys where Stx causes HUS.

**Keywords** amyloid; curli; EAEC; EHEC; haemolytic uraemic syndrome
**Subject Categories** Microbiology, Virology & Host Pathogen Interaction

## Introduction

In 2011, a major outbreak of a highly virulent enterohaemorrhagic *Escherichia coli* (EHEC) of the serotype O104:H4 that caused bloody diarrhoea and an unusual high incidence of haemolytic uraemic syndrome (HUS) occurred in Germany. Overall, 3,842 persons were affected, with 2,987 patients suffering from gastroenteritis and 855 patients developing HUS, with 53 patients dying from these complications (Bielaszewska *et al*, 2011; Muniesa *et al*, 2012). Virulence

gene profiling (Bielaszewska *et al*, 2011) as well as rapid whole-genome sequencing (Brzuszkiewicz *et al*, 2011; Mellmann *et al*, 2011; Rasko *et al*, 2011; Rohde *et al*, 2011) revealed that the outbreak strain, rather than being a classical EHEC, is closely related to enteroaggregative *E. coli* (EAEC), in particular to strain 55989, an isolate from Central Africa causing chronic diarrhoea (Mossoro *et al*, 2002). Not only are the chromosomal genomes highly similar, but among the three plasmids of the outbreak strain, one (pAA) is associated with EAEC-typical aggregative adherence (Harrington *et al*, 2006), although pAA[H104:H4] carries the genes for type I aggregative adherence fimbriae (AAF/I), while pAA[55989] contains the AAF/III gene cluster. Expression of these fimbrial genes is under the control of the EAEC-specific global virulence regulator AggR (Morin *et al*, 2013). In addition, the O104:H4 strain produces chromosomally encoded virulence factors typically associated with EAEC, for example three serine protease autotransporters (SPATE) released from the cell by autocleavage that include Pic, a mucinase that facilitates intestinal colonization (Harrington *et al*, 2009).

On the other hand, the outbreak strain resembles EHEC in producing Shiga toxin (Stx), a potent protein synthesis inhibitor encoded by a *stx2* prophage (Johannes & Romer, 2010), which is inserted at the *wrbA* gene and is of a type commonly found in cattle-associated EHEC strains (Beutin *et al*, 2013). However, the O104:H4 strain lacks a second *stx2* prophage that in many classical EHEC (e.g. in the O157:H7 strain EDL933) is inserted at the *mlrA* (*yehV*) gene (Serra-Moreno *et al*, 2007). Transfer of Stx into the bloodstream leads to kidney damage and HUS (Kaper *et al*, 2004).

With more than 22% of the patients developing HUS, the 2011 O104:H4 outbreak strain seems to have the highest incidence of HUS ever reported for an enterohaemorrhagic *E. coli*. This was tentatively attributed to its strong EAEC-specific aggregative adherence combined with Stx production (Bielaszewska *et al*, 2011). Recently, however, it was reported that the entire pAA plasmid—while being essential for 'stacked brick' aggregative adherence typically observed with EAEC—is dispensable for colonization and inducing diarrhoea in an infant rabbit model, whereas Stx and the above-mentioned SPATE autotransporters are of crucial importance (Munera *et al*, 2014). The O104:H4 outbreak strain is also highly

---

1   Institute of Biology / Microbiology, Humboldt-Universität zu Berlin, Berlin, Germany
2   Institute of Microbiology and Epizootics, Freie Universität Berlin, Berlin, Germany
   *Corresponding author. Tel: +49 2093 8101; Fax: +49 2093 8102; E-mail: Regine.hengge@hu-berlin.de
   †These authors contributed equally to this work

superior to the classical EHEC O157:H7 strain EDL933 in producing thick submerged biofilms *in vitro* (Al Safadi *et al*, 2012). During colonization of germ-free mice, the *in vivo* expression of *aggR* (encoding the EAEC virulence regulator) and *pgaA* (essential to produce the biofilm-associated exopolysaccharide poly-β-1,6-N-acetyl-D-glucosamine or PGA) increased > 1,000-fold, which correlated with a similarly strong activation of *stx2* and several other virulence genes. Maximal induction of Stx developed only after about 2 weeks, whereas *stx2* expression in O157:H7 peaked already after 5–7 days. This suggests that during an infection with O104:H4, a stable biofilm is first established before virulence gene expression is massively induced and is consistent with a delay in developing symptoms of infection in comparison with O157:H7 (Al Safadi *et al*, 2012). Overall, the factors that contribute to adherence, biofilm formation and colonization during an O104:H4 infection clearly require further characterization.

Bacterial adherence and biofilm formation are almost universally stimulated by the second messenger c-di-GMP. In non-pathogenic *E. coli* K-12, this signalling molecule is synthesized by 12 diguanylate cyclases (DGC, containing GGDEF domains) and degraded by 13 phosphodiesterases (PDE, containing EAL domains), which differ by their modes of expression and of activation via their N-terminal sensor domains (Jenal & Malone, 2006; Hengge, 2009; Schirmer & Jenal, 2009). In *E. coli* and related bacteria, c-di-GMP stimulates the production and secretion of PGA by directly binding to the cell envelope-spanning Pga machinery encoded in the *pgaABCD* operon (Wang *et al*, 2004; Itoh *et al*, 2008; Steiner *et al*, 2013). In a cascade involving several DGCs, PDEs, the transcription factor MlrA and RNA polymerase containing the stationary phase sigma factor RpoS ($\sigma^S$), c-di-GMP also stimulates the expression of the transcription factor CsgD (Pesavento *et al*, 2008; Lindenberg *et al*, 2013). This biofilm regulator then activates the production of amyloid curli fibres and cellulose (Römling *et al*, 1998a,b, 2000; Zogaj *et al*, 2001), which encase biofilm-inhabiting cells in a massive and tight matrix network and determine microarchitecture and macroscopic morphology of biofilms (Serra *et al*, 2013a,b; Serra & Hengge, 2014).

The observation that the outbreak O104:H4 strain is an excellent biofilm former, but that on the other hand c-di-GMP signalling has not been studied in EAEC, prompted us to analyse GGDEF/EAL domain-encoding genes in the O104:H4 genome in comparison with other EAEC, EHEC and *E. coli* K-12. In addition, we compared the ability to produce amyloid curli fibres and cellulose, that is the biofilm matrix components underlying the complex biofilm morphogenesis, as well as the cellular levels of the biofilm regulator CsgD in these strains. In summary, we report that the outbreak O104:H4 strain shows a unique combination of features that do not only relate to its specific biofilm properties but can contribute to explain its high virulence.

## Results

### A highly expressed novel diguanylate cyclase (DgcX) and other variations in c-di-GMP signalling genes and proteins in the outbreak *E. coli* O104:H4 and related EAEC

Complete genome sequence surveys as the ones performed so far for the outbreak O104:H4 strain usually focus on the presence or absence of entire genes or longer genomic regions such as prophages, pathogenicity islands or other insertion elements, simply because they have to deal with massive amounts of sequence data. In our study, however, we focus on a specific subset of genes, that is those involved in c-di-GMP signalling and biofilm matrix production, which allowed us to take into account even SNPs and other small genomic variations and not only to verify these by re-sequencing of PCR fragments, but also to experimentally analyse them for functional and regulatory consequences.

Two clonal isolates of the outbreak O104:H4 strain were used in our study, LB226692 (for which the genome sequence is available) and RKI II-2027 (designated as the official outbreak strain). For comparison, we included the closely related EAEC O104:H4 strains 55989 and HUSEC041 as well as the classical EHEC O157:H7 strain EDL933 and the non-pathogenic *E. coli* K-12 strain W3110. As a first step in our analysis, the genome sequences of these strains were searched for genes encoding GGDEF/EAL domain proteins, which act as diguanylate cyclases (DGC) and phosphodiesterases (PDE) that synthesize and degrade the second messenger c-di-GMP.

In the three EAEC strains LB226692, 55989 and HUSEC041, an extra GGDEF gene not present in EDL933 and W3110 was detected and termed *dgcX* (Fig 1A). This gene codes for a DGC of the type that is subject to I-site-mediated product feedback inhibition, since DgcX features an intact A-site (the enzymatically active centre with the GG(D/E)EF motif) as well as an I-site (a secondary c-di-GMP-binding site characterized by an RxxD motif only 5 amino acids upstream of the A-site; Supplementary Fig S1). Besides the GGDEF domain, it contains a hydrophobic N-terminal domain of approximately 270 amino acids predicted to fold into eight transmembrane helices. A similar putative sensor domain is also present in two other GGDEF domain proteins of *E. coli*, that is the DGC YcdT (see below) and YeaI, a protein with a degenerate A-site but intact I-site motif (Supplementary Fig S1).

In order to find out how widespread this novel DGC is among *E. coli* strains, we searched for it in 56 additional publicly available genome sequences which include pathogenic, commensal and probiotic strains of all phylogroups. As expected, the *dgcX* gene was also detected (Fig 1B) in another isolate of the 2011 outbreak strain (2011C-3493) sequenced independently (Grad *et al*, 2013) as well as in closely related O104:H4 strains (isolates 2009EL-2050 and 2009EL-2071) originating in Georgia in 2009 (Miko *et al*, 2013). Furthermore, two enterotoxic *E. coli* strains (ETEC; isolates E24377A and H10407) carry *dgcX* inserted at the same chromosomal position as the O104:H4 strains, that is at the *attB* locus (the primary insertion site for phage lambda), but adjacent to different prophages (or what has remained of these, as the prophage-related sequence in strain E24377A is only 29.9 kbp long). Finally, *dgcX* was also found in the commensal *E. coli* strain SE11, where it is inserted at a different chromosomal location (to the left of the *uspF-ompN* region). Also here, *dgcX* is flanked by a 39.3-kbp prophage region that substitutes for a region of approximately 11.3 kbp in the *E. coli* K-12 genome that includes Rac prophage genes as well as *trkG* (encoding a potassium transporter) (Fig 1C). The localization of *dgcX* always directly adjacent to lambdoid prophage DNA, which in most—but not all—cases is inserted at *attB*, suggests that *dgcX* can 'hitchhike' on these prophages by specialized transduction. Consistent with such horizontal transfer and despite its relatively rare occurence (52 out of 61 *E. coli* genomes analysed did not

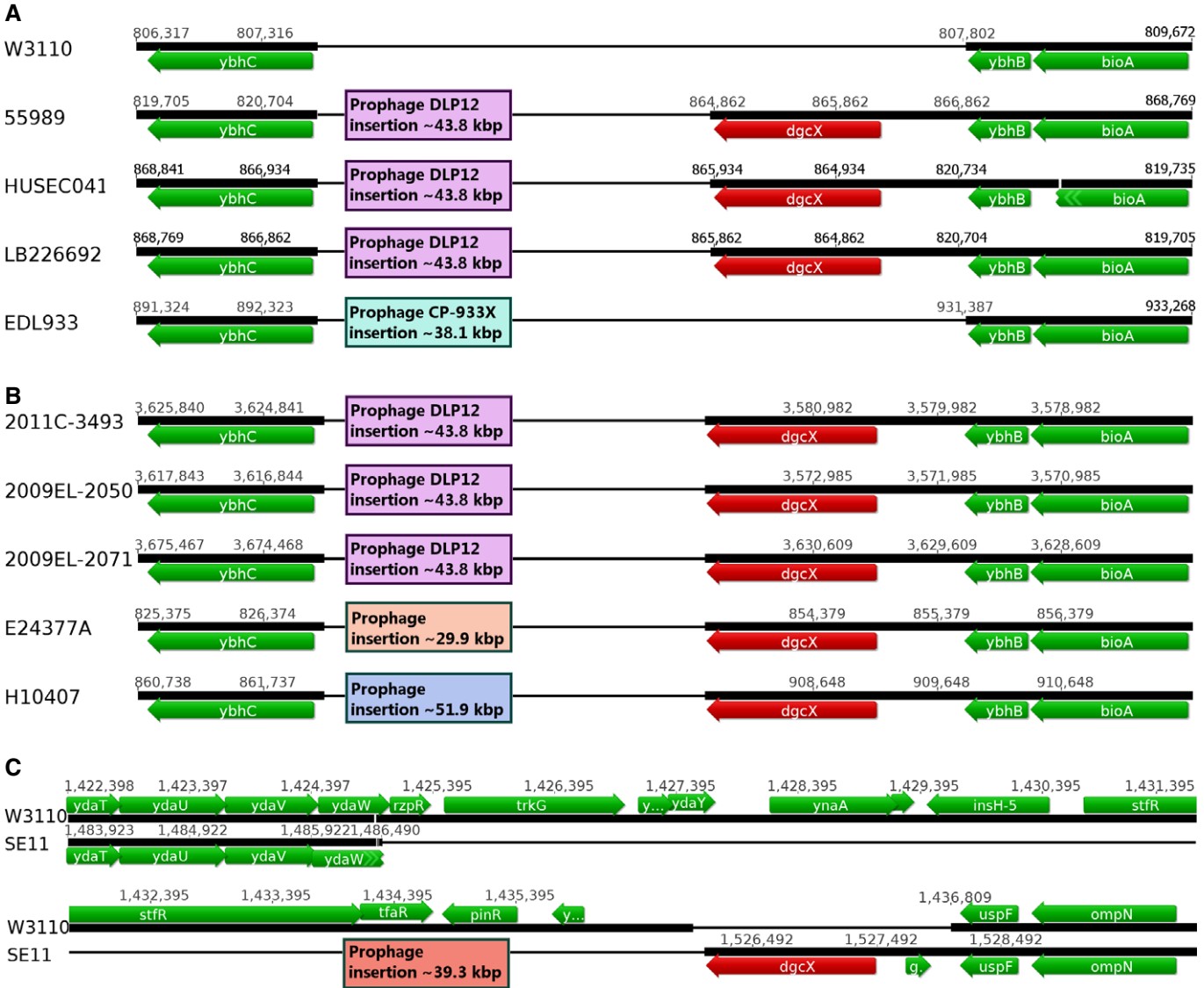

**Figure 1.  A novel diguanylate cyclase-encoding GGDEF gene, *dgcX*, in the O104:H4 outbreak strain and related EAEC in comparison with *Escherichia coli* K-12 W3110 and classical EHEC EDL933.**

A  Genomic regions with insertion of *dgcX* and adjacently inserted prophage gene clusters (boxes) are shown for the five strains under experimental study here.

B  Genomic regions of additional strains that carry *dgcX* at the same chromosomal site as the strains shown in (A).

C  The commensal *E. coli* strain SE11, which shows a prophage/*dgcX* insertion at another chromosomal position, is shown in comparison with *E. coli* K-12 W3110, which features a different set of genes at this position.

Data information: Open reading frames and their direction of transcription are indicated by arrows, with red arrows indicating the DGC gene *dgcX*. Thick or thin horizontal lines indicate the presence or absence of the corresponding sequences in the genomes of the indicated strains, respectively.

contain the gene), *dgcX* is not strictly associated with a specific pathotype or a single phylogroup (ETEC H10407 belongs to phylogroup A, whereas the other *dgcX*-carrying *E. coli* strains belong to phylogroup B1 (White *et al*, 2011)). Nevertheless, it seems prevalent in EAEC O104:H4 strains (no matter whether positive or negative for Stx) that belong to phylogroup B1.

Searching for additional variations in GGDEF/EAL genes, we found that the three EAEC O104:H4 strains as well as EHEC O157:H7 also contain an extended version of the DGC gene *yneF*, which revealed that in *E. coli* K-12 this gene is 5′-truncated by a deletion

(Fig 2A). This has led to start codon misannotation and explains why *yneF* is not expressed in strain W3110 (Sommerfeldt *et al*, 2009). Full-size YneF as encoded by the pathogenic strains contains an N-terminal highly hydrophobic domain of approximately 300 amino acids not found in any other *E. coli* protein, which is linked to a GGDEF domain with intact A-site but no I-site, suggesting YneF is a DGC not subject to product feedback inhibition.

Furthermore, the DGC gene *ycdT*, which was suggested to play a role in the production of the exopolysaccharide PGA (Jonas *et al*, 2008) since it is transcribed divergently from the *pgaABCD* operon

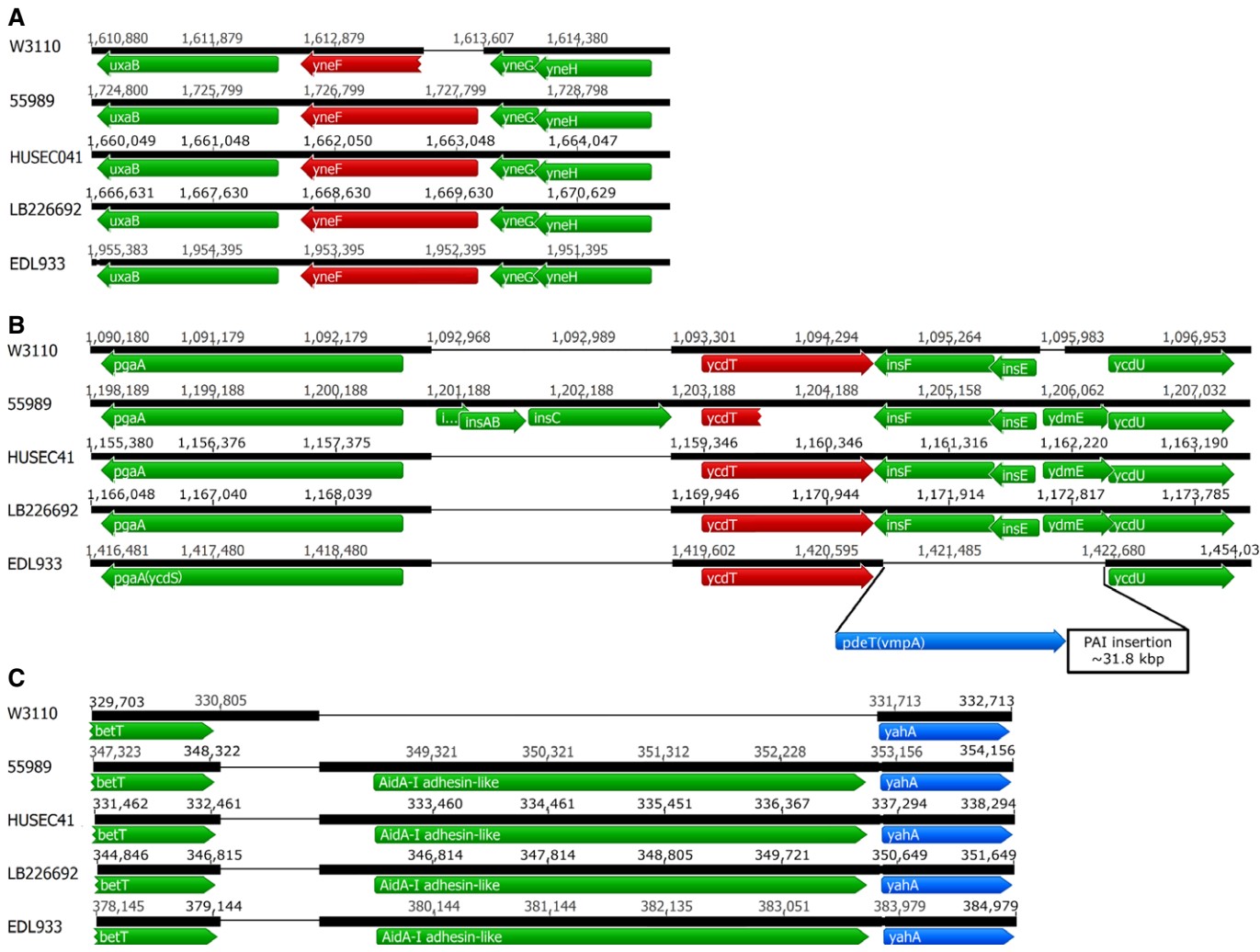

**Figure 2. Genomic context of several GGDEF/EAL genes encoding DGCs and PDEs showing alterations in the O104:H4 outbreak strain and related EAEC in comparison with *Escherichia coli* K-12 W3110 and classical EHEC EDL933.**

A–C  Open reading frames and other features are represented as in Fig 1. Red and blue arrows indicate DGC and PDE genes, respectively.

(Fig 2B), showed an interesting variation among the EAEC strains. While the *pgaA-ycdT* region of the O104:H4 outbreak strain and of HUSEC041 corresponds to that of the K-12 strain W3110 (Fig 2B; Supplementary Fig S2), with *ycdT* being intact, strain 55989 carries a 7-bp insertion in *ycdT* that generates a translational frameshift that results in a truncated N-terminal protein fragment missing part of the sensor domain and the entire GGDEF domain of YcdT. Moreover, this corrupted *ycdT* gene is separated from the *pgaABCD* operon by an intervening IS1 element (Fig 2B), which also disrupts the *pgaA* promoter sequence (as determined in *E. coli* K-12 by Wang *et al* (2005), with a K-12-like sequence confirmed here for W3110, the outbreak strain and HUSEC041). IS1 actually not only contains a strong -35-like region very close to its end that can combine to the original -10 element still present upstream of *pgaA* to yield a new promoter, but also carries a full promoter region slightly further upstream (Supplementary Fig S2). Both promoter elements are directed outwards, that is towards the *pga* operon. Thus, insertion of the end of IS1 upstream of *pgaABCD* creates a novel double

promoter region expected to result in stronger expression of the *pga* genes in strain 55989 than in strains without this IS1 insertion—a situation actually observed earlier with IS1 insertions upstream of other genes (Prentki *et al*, 1986; Barker *et al*, 2004). Notably, in EHEC EDL933, the divergently transcribed *pgaABCD-ycdT* region is intact, and in fact, an extra PDE-encoding gene (*vmpA/pdeT*; EDL933_Z1528) (Branchu *et al*, 2012) not present in the other strains under study here is inserted immediately downstream of *ycdT* (Fig 2B).

A fourth DGC gene altered in the three EAEC strains is *yedQ* encoding an active DGC with a large periplasmic sensor domain of > 300 amino acids, which is disrupted by a stop codon in the part specifying the periplasmic domain (Supplementary Table S1). However, already the second codon after this stop codon is an ATG, which by translational coupling is highly likely to trigger a translational restart (if readthrough does not occur anyway). This would generate a N-terminally truncated YedQ protein with nevertheless intact GGDEF domain, which may or may not function as a DGC.

Also the gene encoding the PDE YahA is affected by a genetic rearrangement in the EAEC strains as well as in the EHEC O157:H7 strain (Fig 2C). Here, the promoter region of *yahA* as present in *E. coli* K-12 is replaced by a new long open reading frame which encodes an uncharacterized AidA-I adhesin-like autotransporter. The insertion of this gene (annotated as EC55989-0317 in strain 55989), which is actually followed by a nucleotide sequence able to fold into a hairpin structure, may silence the expression of *yahA*. Alternatively, the two genes may be co-expressed from a new bicistronic operon under unknown conditions.

Any other GGDEF/EAL genes of the EAEC O104:H4 outbreak strain or its close EAEC relatives were not significantly different in comparison with W3110 (Supplementary Table S1; occasional synonymous codons or codons resulting in conservative amino acid exchanges were not taken into account). Taken together, our genomic analyses indicate an enhanced c-di-GMP accumulation potential in all three EAEC strains, which is based on two extra DGCs, that is DgcX and YneF, and a potentially reduced expression of the PDE YahA. In strain 55989, DgcX may also functionally compensate for the absence of the DGC YcdT, since the two proteins are direct paralogs sharing not only the GGDEF domain but also a N-terminal sensor domain of unknown function.

Finally, we experimentally analysed the expression of the two extra DGC genes *dgcX* and *yneF* from EAEC by generating single-copy *lacZ* reporter fusions inserted into the chromosome of strain W3110 in the very same way as the *lacZ* reporter fusions to all other GGDEF/EAL genes previously described (Sommerfeldt *et al*, 2009), thus allowing a direct comparison of their expression. Strikingly, the *dgcX* gene showed far higher expression than any other known DGC gene of *E. coli* (Fig 3). *dgcX* seems to be transcribed by the vegetative as well as by stationary phase RNA polymerase (containing sigma subunits RpoD and RpoS, respectively), since it was already strongly expressed in growing cells, but was further induced during entry into stationary phase in a RpoS-dependent manner (Supplementary Fig S3). By contrast, *yneF* is a very weakly active DGC gene with mainly RpoS-dependent expression (Fig 3; Supplementary Fig S3).

### Morphology of macrocolony biofilms and production of amyloid curli fibres differ between the O104:H4 outbreak strain and closely related EAEC strains

Bacterial macrocolonies grown for extended times on agar plates represent biofilms that can produce complex morphological patterns that have been termed 'wrinkled', 'rugose' or 'rdar' (for <u>r</u>ed, <u>d</u>ry <u>a</u>nd <u>r</u>ough on agar plates containing Congo red (CR); Fig 4). In enteric bacteria, these patterns depend on the formation of amyloid curli fibres and cellulose, which are stained with CR (Römling, 2005). While concomitant production of curli fibres and cellulose generates network-like morphological patterns with high ridges interlinked by smaller wrinkles, strains like *E. coli* K-12 W3110, which produce high amounts of curli fibres but no cellulose, grow in a pattern of concentric wrinkled rings that arise by breakage of the non-elastic curli-only matrix surrounding the starving cells in the upper layer of the macrocolony (Serra *et al*, 2013a,b; Serra & Hengge, 2014). Biosynthesis of both biofilm matrix components, which in most *E. coli* strains occurs below 30°C only and is stimulated by low salt concentrations, depends on a regulatory cascade consisting of the stationary phase sigma subunit of RNA polymerase,

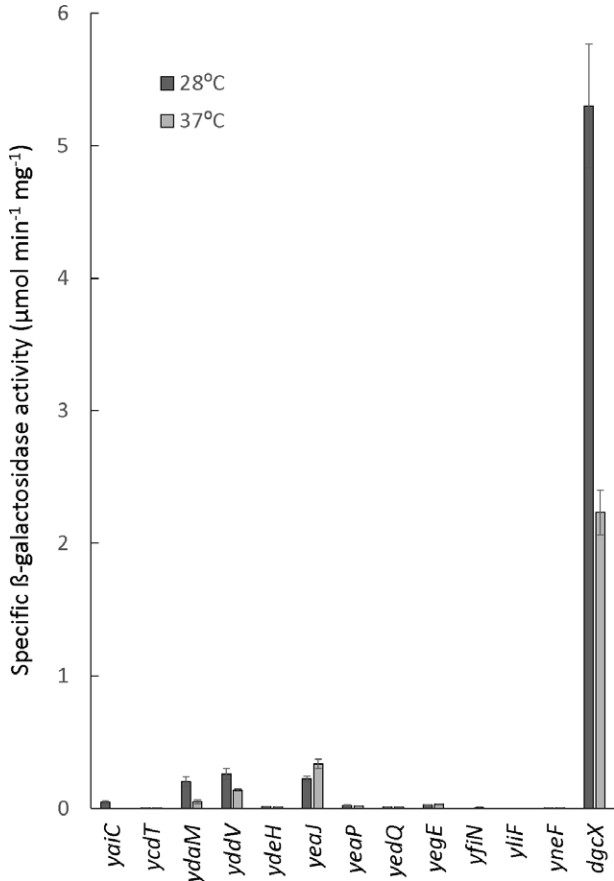

**Figure 3. Expression of *lacZ* reporter fusions to *dgcX* and other *Escherichia coli* DGC genes.**
Derivatives of strain W3110 carrying single-copy *lacZ* fusions in *dgcX* and *yneF* (relevant sequences obtained from the O104:H4 outbreak strain) or in all other DGC genes previously described in *E. coli* K-12 (Sommerfeldt *et al*, 2009) were grown in LB at 28 and 37°C, and specific β-galactosidase activities were determined in overnight cultures.

RpoS ($\sigma^S$), several DGCs and the two transcription factors MlrA and CsgD (Hengge, 2010; Lindenberg *et al*, 2013).

Like strain W3110, the outbreak O104:H4 strain produced a ring-like macrocolony pattern, whereas the EAEC strains 55989 and HUSEC041 generated delicate network-like structures (Fig 4A; for colony morphology at different temperatures and also in the presence of salt, see Supplementary Fig S4). This clear difference indicates that strains 55989 and HUSEC041 produce curli fibres as well as cellulose, whereas the outbreak strain synthesizes high amounts of curli fibres, but is negative for cellulose. By contrast, the O157:H7 strain EDL933 grew into flat and unstructured macrocolonies, consistent with its second Stx2-producing prophage being inserted in the *mlrA* gene (Supplementary Fig S5) (Shaikh & Tarr, 2003; Serra-Moreno *et al*, 2007; Uhlich *et al*, 2013), which encodes the transcription factor that is essential for the expression of CsgD, which in turn is required to produce curli and cellulose.

Adding CR to the agar plates produced the dark red staining characteristic for high-curli production, while not affecting the network-like or ring-structured macrocolony morphologies (Fig 4B). In addition, all EAEC strains, including the outbreak strain, were observed

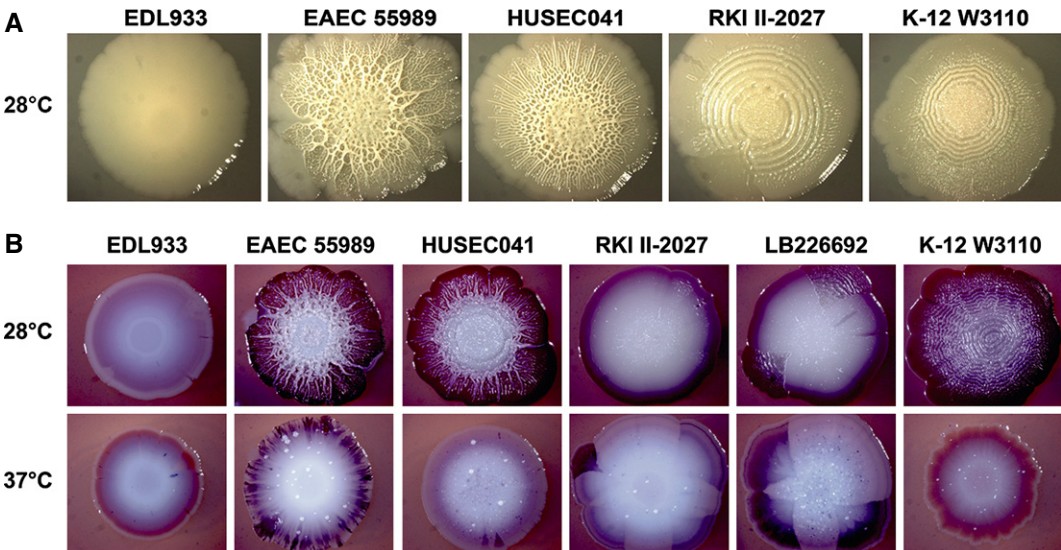

**Figure 4. Macrocolony morphology of the outbreak O104:H4 strain in comparison with 55989, HUSEC041, EDL933 and W3110.**

A  The strains were spotted onto salt-free LB agar plates and grown at 28°C for 7 days.

B  Similar to (A), but the agar plates also contained Congo red and were grown at the indicated temperatures for 7 days.

to produce a thick layer of a whitish, non-CR-staining material on their macrocolony surfaces that may represent an unknown additional biofilm matrix component. Also W3110 produced some of this material, but not enough to cover and affect the overall macrocolony structure. Moreover, an interesting differential behaviour was observed in macrocolonies grown with CR at 37°C (Fig 4B). While strain 55989 generated an almost regular pattern of numerous small dark red sectors, HUSEC041 produced such sectors only rarely and the outbreak O104:H4 strain usually generated a few large dark red sectors. In other words, the EAEC strains produce curli fibres also in macrocolonies grown at 37°C, but do so with different spatial patterns of distinct subpopulations. By contrast, macrocolonies of K-12 strain W3110 (curli-negative at 37°C) and EHEC EDL933 (curli-negative at 28 and 37°C) generate a circular pattern of light brown-red colour when grown at 37°C that may reflect production of PGA, which also stains with CR (Izano *et al*, 2008).

### In contrast to related EAEC, the outbreak O104:H4 strain does not produce cellulose

Its macrocolony morphology (Fig 4A) indicated that the O104:H4 strain is unable to generate cellulose. This was confirmed by its reduced staining with calcofluor, similar to what was observed for W3110 (Fig 5). By contrast, EAEC 55989 and, in particular, HUSEC041 showed stronger calcofluor staining typical for high cellulose production (Serra *et al*, 2013a). EHEC EDL933 was completely devoid of staining, consistent with an absence of both cellulose and curli (also curli stains with calcofluor, although to a lesser extent than cellulose).

In search for an explanation for the outbreak strain's inability to produce cellulose, we analysed the *bcs* genes involved in cellulose biosynthesis (Supplementary Fig S6A). The *yhjR-bcsQABZC* operon, which contains the genes for cellulose synthase (*bcsA* and *bcsB*) and several accessory factors, is identical in the outbreak strain and its

closely related EAEC strains. However, the outbreak strain contains a unique insertion of a C in codon 448 of *bcsE* (the presence of four instead of only three C at this position was confirmed by resequencing of a PCR fragment). This generates a synonymous codon for proline (now CCC instead of CCA), but shifts the reading frame of the downstream coding sequence by +1. This would result in a C-terminally truncated BcsE protein that misses the last 78 amino acids, which includes a region highly conserved among BcsE proteins of different species. A *bcsE* mutation has been shown to affect cellulose biosynthesis in *Salmonella* (Solano *et al*, 2002). In addition, we generated mutants with non-polar deletions in *bcsE*, *bcsF* and *bcsG* in the *E. coli* K-12 strain AR3110 (a cellulose⁺ derivative of W3110 (Serra *et al*, 2013a)). In contrast to the parental strain, these mutants indeed showed a ring-like macrocolony morphology typical for curli-only producers (Supplementary Fig S6B), indicating that not only the *bcsE* mutation in the outbreak strain is most likely responsible for its cellulose-negative phenotype, but that all three genes in the *bcsEFG* operon are involved in cellulose biosynthesis and/or secretion.

### Altered regulation of expression of the biofilm regulator CsgD in the O104:H4 outbreak strain

Biosynthesis of curli directly follows the expression of the regulator CsgD, which is essential to activate transcription of the *csgBAC* operon. The observation that EAEC 55989 as well as the outbreak strain shows partial dark red staining on CR plates also at 37°C (Fig 4B) suggested that the strict temperature regulation of CsgD—as found in *E. coli* K-12 and other strains—may be relaxed in these EAEC. We therefore quantified CsgD protein in LB-grown macrocolonies by immunoblotting (Fig 6). As expected, all strains except EHEC O157:H7 had high levels of CsgD when grown at 28°C and in the absence of salt (up to approximately 6,000 molecules per cell; see legend to Fig 6 for details of quantification). However, EAEC

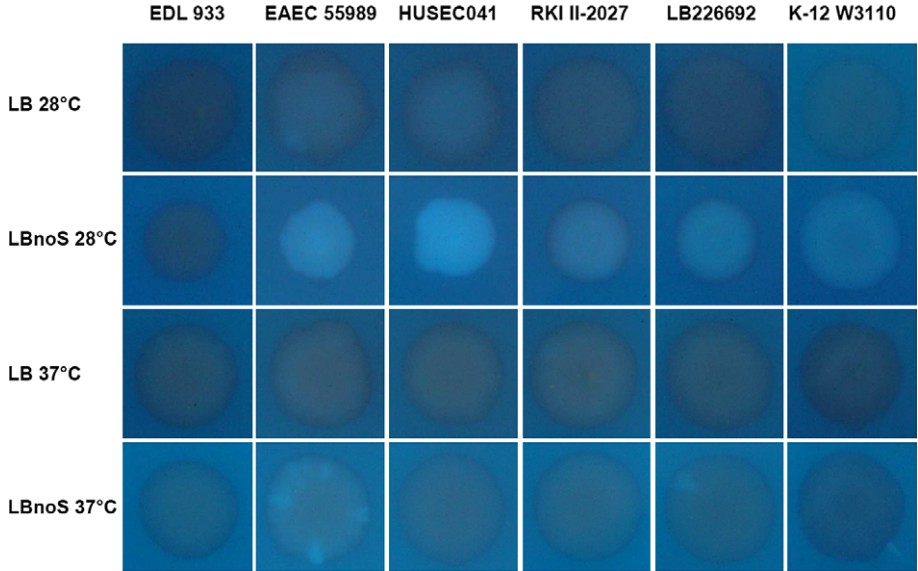

**Figure 5. Cellulose formation of the outbreak O104:H4 strain in comparison with 55989, HUSEC041, EDL933 and W3110.**
The strains were spotted onto standard LB or no-salt LB agar plates also containing calcofluor and grown for 3 days. Cellulose formation was visualized by calcofluor fluorescence.

55989 as well as the outbreak strain displayed high CsgD content also at 37°C and even in the presence of salt. We conclude that temperature and salt regulation of CsgD is relaxed in these strains. A closer inspection of the sequence of the *csgD* promoter region in these strains did not reveal any variations in comparison with W3110 and HUSEC041, suggesting that unknown regulatory mutations acting *in trans* are responsible for this relaxed control.

It should also be noted that these CsgD levels determined in entire macrocolonies represent average values and that high-curli expression at 37°C occurred in sectors as revealed in the presence of CR (Fig 4B). With the many small sectors observed with EAEC 55989, this led to highly reproducible average CsgD levels. The O104:H4 outbreak strain, however, generated variable low numbers of large high-curli sectors and therefore also variable average CsgD levels in different macrocolonies. We therefore took samples for immunoblotting from distinct regions of 10 independent macrocolonies of the outbreak strain grown in the presence of CR, that is: (i) from the centre; (ii) from peripheral regions not stained with CR; and (iii) from red sectors of different colour intensities (Fig 7A). While central regions of the macrocolonies contained similar medium CsgD levels, the CR-stained sectors in the peripheral zones showed clearly higher CsgD content than adjacent non-stained regions (Fig 7B). We conclude that the outbreak strain has the potential to produce high levels of CsgD and therefore curli fibres at 37°C, but does so in subpopulations that arise spontaneously at high frequency.

### Frequent spontaneous occurrence of high-curli derivatives and mutation rates of the O104:H4 outbreak strain

We then asked whether the frequent dark red sectors with increased curli production and CsgD content in macrocolonies of the outbreak strain represent a stable phenotype indicative of spontaneously arising mutations. Regrowing cells taken directly from a dark red sector

of a primary macrocolony generated secondary macrocolonies (again at 37°C) with generally darker CR staining that now occasionally produced white sectors (Fig 8A-1). When the inoculum for the secondary macrocolony was taken from an already white sector of a primary macrocolony, also this phenotype was reproduced in the entire secondary macrocolony (Fig 8A-3). Interestingly, a non-sector region of a primary macrocolony produced secondary macrocolonies with streaks of strongly curli-producing cells (Fig 8A-2), indicating high variability in the already confluent growth zones of the primary colonies where visible sectors could no longer form.

For a second series of experiments, cells taken from dark red sectors of eight independent primary macrocolonies (grown at 37°C) were purified as single cell colonies. One derivative of each was grown again into a macrocolony, but now at 28°C in order to see potentially altered macrocolonial morphology of the derivatives (Fig 8B). Along with increased CR staining, three new morphotypes were observed in comparison with the original outbreak strain: (i) type 1 produced a rather drastic thick ring structure (Fig 8B, isolate 4); (ii) type 2 showed a finer ring structure that looked similar to that of sectors already observed with primary macrocolonies of the outbreak strain (isolates 2 and 3; compare to Fig 4B); and (iii) type 3 displayed rather weak and wrinkly rings (isolates 5, 6 and 8). Two isolates (1 and 7) were considered not significantly different from the original outbreak strain, indicating that CR-staining red sectors still contain unaltered original cells. Sequencing of the *csgD* promoter region did not reveal any mutations in these eight independent isolates. It should also be noted that—based on their macrocolony morphology—none of these 'red sector derivatives' had regained the ability to generate cellulose.

Overall, we conclude from these data that: (i) the O104:H4 outbreak strain frequently generates mutants with increased CsgD and curli content also at 37°C; (ii) this phenotype is stably maintained and therefore is most likely due to spontaneous mutations; (iii) based on distinct macrocolony morphology, different types of

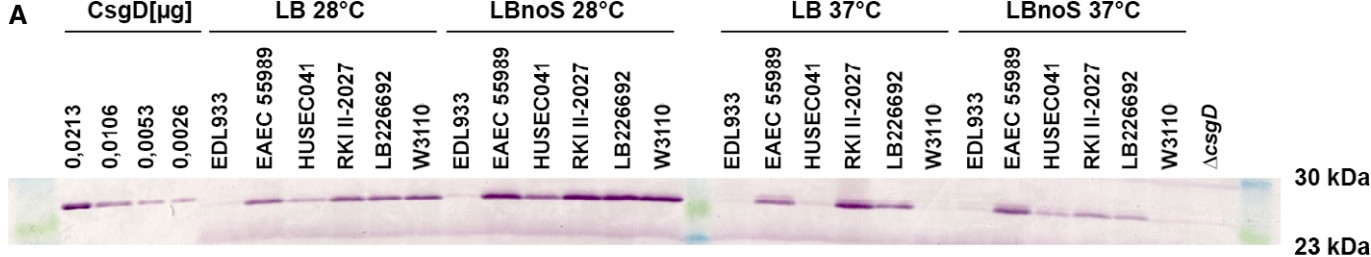

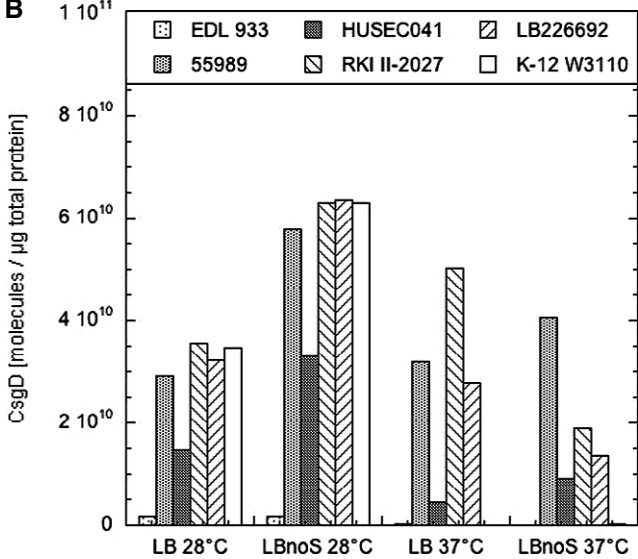

**Figure 6.  Cellular levels of the biofilm regulator CsgD in macrocolonies of the outbreak O104:H4 strain in comparison with 55989, HUSEC041, EDL933 and W3110.**

A   For the determination of cellular CsgD levels by immunoblotting, the strains were spotted onto standard LB or no-salt LB agar plates and grown at 28 and 37°C for 5 days. Purified CsgD protein (of slightly larger size due to the presence of a His6-tag) was used in the indicated amounts as a standard for quantifying the CsgD content in the cellular samples.

B   Quantification of cellular CsgD content. Based on the dimensions of early stationary phase cells of *Escherichia coli* K-12 and cell number/optical density ratios (Lange & Hengge-Aronis, 1991) and a standard cellular protein/optical density ratio (Miller, 1972), $10^{10}$ molecules/µg cellular protein corresponds to approximately 1,000 molecules/cell.

such mutations can occur; and (iv) these mutations are not located *in cis*, that is the *csgD* promoter region, but affect regulatory components that act *in trans* on CsgD expression.

These findings also raised the question of the general mutation rate of the O104:H4 outbreak strain, which was compared for all strains under study here as the frequency of spontaneously arising rifampicin-resistant mutants. The outbreak strain generated such mutants at a frequency that was about threefold, fivefold and tenfold higher than the frequencies measured for W3110, EAEC 55989 and EHEC O157:H7, respectively (Fig 9). Interestingly, however, EAEC HUSEC041 showed a similar high general mutation rate as the outbreak stain, although it did not frequently generate CR-staining sectors with high CsgD and curli expression. This suggests that the frequent occurrence of 'red sector derivatives' of the outbreak strain is not just a consequence of its relatively high general mutation rate, but may be due to processes more specific to the regulation of CsgD. Consistent with this interpretation, a sequence comparison of well-characterized 'mutator' genes, which encode DNA error prevention and repair systems (Chopra *et al,*

2003), did not reveal any variations among EAEC 55989, HUSEC041 and the outbreak strain (Supplementary Table S2) despite their different sectoring behaviour.

## Discussion

Biofilm-related features such as strong adherence and production of the biofilm matrix exopolysaccharide PGA in combination with strong Stx expression were proposed to play a role in the pronounced virulence of the 2011 O104:H4 outbreak strain (Bielaszewska *et al*, 2011; Mellmann *et al*, 2011; Al Safadi *et al*, 2012). In general, adherence and biofilm matrix production are promoted by the second messenger c-di-GMP (Jenal & Malone, 2006; Hengge, 2009). This study indeed revealed clear differences in genes related to c-di-GMP signalling in the O104:H4 outbreak strain as well as the closely related EAEC strains 55989 and HUSEC041 in comparison with other *E. coli* strains. The most striking difference is the presence of DgcX, a novel DGC with higher expression than any

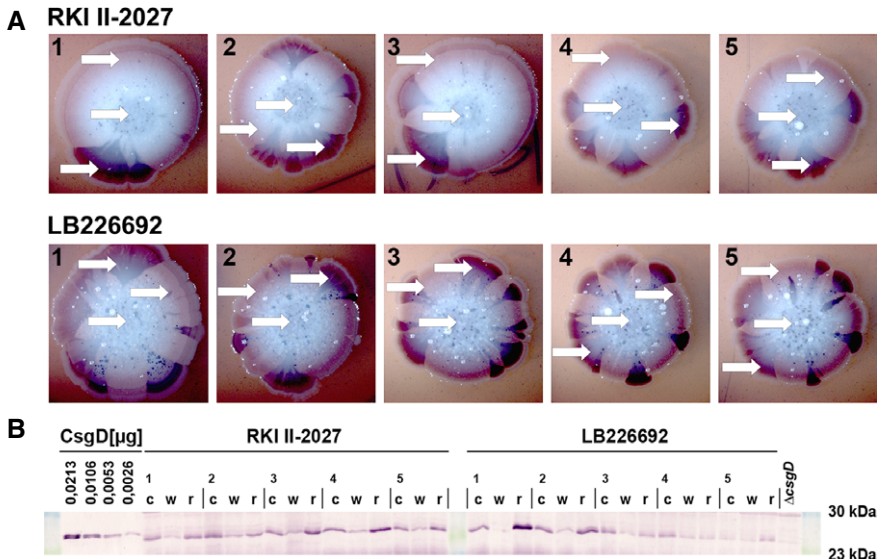

**Figure 7.  Macrocolonies of the outbreak O104:H4 strain highly frequently generate sectors of cells that stain dark red on Congo red plates and contain high levels of CsgD.**

A   The two isolates of the outbreak strain were spotted onto Congo red indicator plates and grown at 37°C for 5 days.

B   Cells were directly taken (arrows in A) from large red sectors (r) as well as from the centre (c) and non-sectored 'white' areas in the outer zones (w) of the colonies, and relative CsgD levels were determined by immunoblotting (the order of samples corresponds to that of the photographs in A).

other DGC described so far in *E. coli*. Vegetative as well as stationary phase-specific RNA polymerase (with RpoD and RpoS as sigma factors, respectively) contributes to this strong expression of DgcX, which means that virtually all cells in an EAEC biofilm, no matter whether they inhabit zones where nutrients are provided or not (Serra *et al*, 2013a,b), express DgcX. The activity of DgcX may be controlled by an unknown and possibly host-associated signal via its N-terminal cytoplasmic membrane-inserted sensor domain. A similar input domain occurs in two other GGDEF proteins of *E. coli*, YeaI and YcdT, with the latter probably promoting the synthesis of the biofilm exopolysaccharide PGA (Jonas *et al*, 2008). Remarkably, however, the 'blockbuster' expression of *dgcX* is in sharp contrast to the very weak expression of *ycdT* (Fig 3). Thus, the DGC DcgX is likely to be both highly expressed and highly activated within the host. All this indicates that the outbreak strain and EAEC in general produce high levels of c-di-GMP and lead a biofilm existence in the host, consistent with the *pga* genes being highly expressed in germ-free mice 13–15 days after infection with the O104:H4 outbreak strain (Al Safadi *et al*, 2012).

Other EAEC-typical features may further contribute to high levels of c-di-GMP, including the presence of YneF, a weekly expressed DGC and the insertion of a novel gene right upstream of *yahA* (Fig 2C), which may interfere with expression of the PDE YahA. Also, that this novel intervening gene codes for an AidA-I-like adhesin may be more than a coincidence. This class of autotransporter adhesin (Klemm *et al*, 2004) has been implicated in bacterial aggregation, biofilm formation, adhesion to and invasion of epithelial cells (Wells *et al*, 2010). Moreover, in 7% of the patients, the O104:H4 outbreak strain had lost the pAA virulence plasmid—encoding EAEC-specific aggregative fimbriae—but patients nevertheless developed diarrhoea (Zhang *et al*, 2013). pAA was also found to be dispensable for intestinal colonization and inducing diarrhoea in

infant rabbits infected with the O104:H4 outbreak strain (Munera *et al*, 2014), suggesting that other adhesins also play a crucial role in adherence *in vivo*. Future studies should address a potential involvement of the novel AidA-I-like adhesin observed here.

A major target for positive regulation by c-di-GMP in *E. coli* is the expression of the biofilm regulator CsgD, which is essential for the production of amyloid curli fibres and the exopolysaccharide cellulose in the biofilm matrix (Weber *et al*, 2006; Pesavento *et al*, 2008; Lindenberg *et al*, 2013). Curli fibres and cellulose are not considered virulence factors, because the genes involved in their biosynthesis belong to the core genome of *E. coli*, and in many *E. coli* strains, these matrix components are produced below 30°C only (Bokranz *et al*, 2005). However, the O104:H4 outbreak strain as well as its close relative EAEC 55989 show high CsgD levels and strong curli production also when grown at 37°C and in the presence of salt (Figs 4 and 6), suggesting that CsgD control as operating in many *E. coli* strains is relaxed in the outbreak strain and closely related EAEC.

Specifically in the outbreak strain, strong CsgD and curli expression at 37°C occurs in highly frequently arising large sectors of macrocolonies, which represents a stable phenotype (Fig 8). In fact, the outbreak strain has a relatively high general mutation rate (Fig 9), as has been observed also with some other pathogenic *E. coli* strains, in particular when *E. coli* grows in biofilms (Denamur *et al*, 2002; Wirth *et al*, 2006; Ponciano *et al*, 2009). However, the frequent occurrence of 'high-curli sector derivatives' of the outbreak strain is not just a consequence of its high general mutation rate, since the closely related HUSEC041 has a similar high mutation rate but does not generate high-curli derivatives and shows stable temperature and salt regulation of CsgD. Also, none of the well-known mutator genes are specifically altered in the outbreak strain. We therefore favour the possibility that the

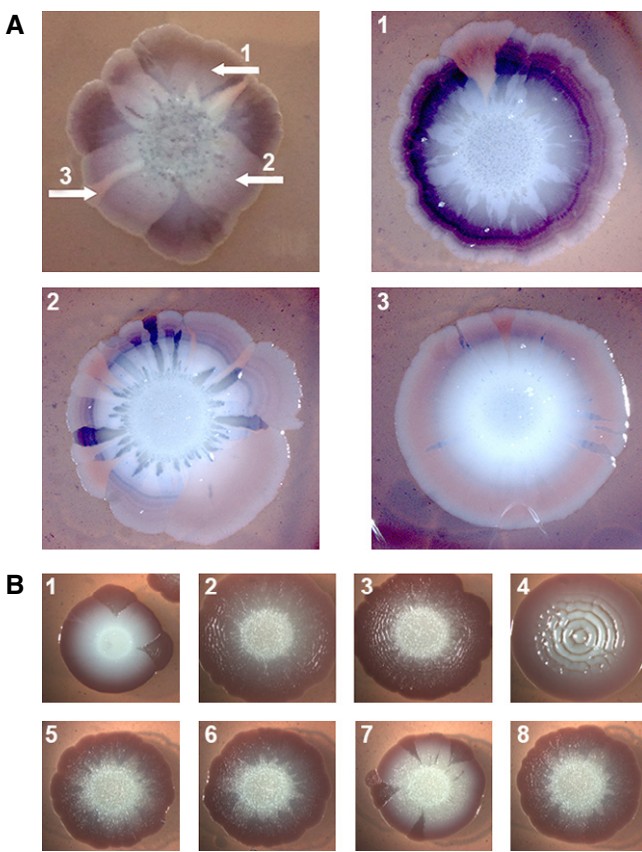

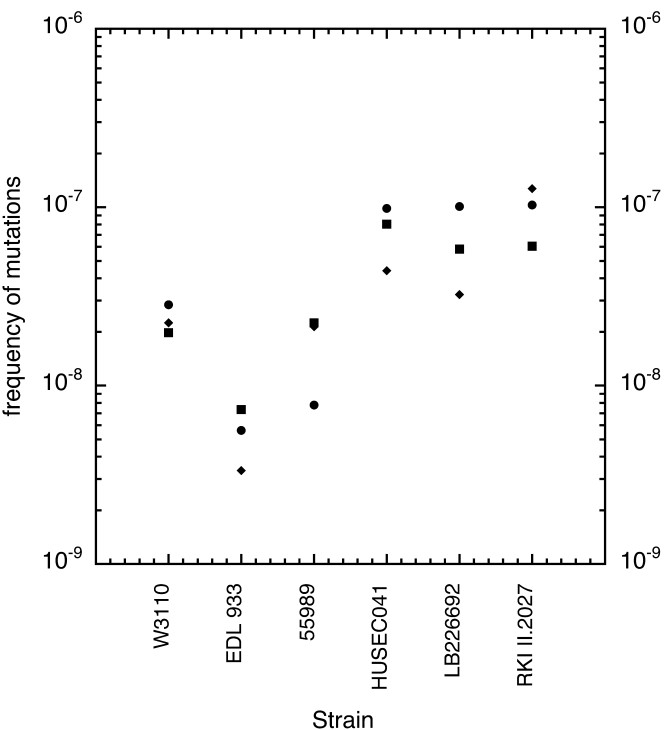

**Figure 9.    Spontaneous mutation rates of the outbreak O104:H4 strain in comparison with 55989, HUSEC041, EDL933 and W3110.**

Mutation frequencies are given as the frequencies of spontaneously arising rifampicin-resistant colonies normalized for the total colony-forming units present in overnight cultures of the strains indicated. Each data point represents the average of three samples taken from a single culture. Circle, diamond and square symbols represent separate cultures.

**Figure 8.    Congo red staining and macrocolony morphotypes of cells in frequently arising sectors in macrocolonies of the outbreak O104:H4 strain are stable traits.**

A   Cells were taken from the indicated areas of the primary macrocolony of the outbreak strain (grown at 37°C on salt-free LB with Congo red for 5 days), resuspended in LB medium, spotted directly onto similar agar plates and grown at 37°C into secondary macrocolonies (numbers of the images correspond to the sampling regions in the primary macrocolony).

B   Purified strains derived from dark red sectors of eight independent primary macrocolonies (grown at 37°C on salt-free LB with Congo red for 5 days) were spotted onto similar agar plates and grown into macrocolonies at 28°C for 5 days.

high-curli derivatives of the outbreak strain carry mutations that affect the control of CsgD expression more specifically.

Conversion to high-curli production has also been observed for certain EHEC O157:H7 strains, where this has been linked to mutations in the *csgD* promoter (Uhlich *et al*, 2001, 2002; Carter *et al*, 2011). However, none of the 'high-curli sector derivatives' of the O104:H4 outbreak strain carried any mutation in > 700 bp upstream of *csgD*; that is, the high-curli phenotype is due to regulatory mutations acting *in trans*. Notably, high-frequency switching between curli⁺ and curli⁻ variants, which was not based on *csgD* promoter mutations, was also described for another EHEC O157:H7 strain, in particular under conditions of nutrient starvation (Carter *et al*, 2011). The complexity of *csgD* regulation (Gerstel *et al*, 2003; Ogasawara *et al*, 2010; Mika & Hengge, 2013) provides for a plethora of candidate genes in which such regulatory mutations may occur. At least three distinct macrocolony morphotypes (at 28°C)

observed for purified 'sector derivatives' (Fig 8B) also suggest curli-enhancing mutations of more than one type. Overall, all this hints to a yet uncharacterized mechanism in distinct *E. coli* strains that provides for rapid phenotypic switching of expression levels of CsgD and therefore high-curli fibre production at 37°C.

The presence of this switching to high-curli production at 37°C in several types of pathogenic *E. coli* strongly points to a function of curli fibres also in the host. Besides their role as a major aggregative and adhesive matrix component in environmental biofilms (Römling *et al*, 1998b; Jonas *et al*, 2007; Serra *et al*, 2013a,b), amyloid curli fibres have been shown to bind to fibronectin and laminin and promote adherence to and even can trigger invasion into epithelial cells (Olsén *et al*, 1989, 1998; Gophna *et al*, 2001, 2002; Rosser *et al*, 2008; Saldana *et al*, 2009). Moreover, by acting on Toll-like receptors 1 and 2 and thereby stimulating production of cytokines such as IL-8, curli fibres are strongly pro-inflammatory (Bian *et al*, 2000; Tükel *et al*, 2005, 2009, 2010). If this occurs in combination with Stx production, the consequences may be severe as proposed already a few years ago for a sorbitol-fermenting Stx-producing *E. coli* O157:NM that also exhibited high-curli expression at 37°C (Rosser *et al*, 2008). Thus, enhanced inflammation and therefore aggravated tissue damage are expected to promote more efficient transition of Stx into the blood stream and thereby increase the probability to develop HUS. Due to its high and deregulated curli production at

37°C, this mechanism is highly likely also for the Stx-producing O104:H4 outbreak strain.

In addition, the outbreak strain displays yet another feature that may contribute to this scenario—its inability to produce cellulose. While its genes for cellulose synthase are intact, the outbreak strain carries a frameshift mutation in *bcsE*, which encodes an accessory protein required for cellulose production. Remarkably, adhesive and pro-inflammatory properties of curli fibres are counteracted by the concomitant production of cellulose (Wang *et al*, 2006; Kai-Larsen *et al*, 2010), which in tight association with curli fibres forms a composite material surrounding the producing cells in the matrix of a biofilm (Serra *et al*, 2013a; Serra & Hengge, 2014). In contrast to the closely related Stx-producing EAEC strain HUSEC041, which shows low CsgD and curli synthesis at 37°C and produces cellulose (Figs 4–6), the outbreak strain thus generates large amounts of cellulose-free and therefore highly inflammatory 'naked' curli fibres at human body temperature.

Taken together, the O104:H4 outbreak strain shows a unique combination of biofilm-related properties not observed in any pathogenic *E. coli* before, including a high c-di-GMP accumulation potential due to an extremely strongly expressed extra DGC (DgcX) and high CsgD and curli expression but no cellulose production at 37°C. These properties can contribute to enhance adhesion and colonization as well as the inflammatory response, which together with production of Stx and SPATEs such as the mucinase Pic combine to high virulence. These properties also point to potential strategies for dealing with future infections with strains related to the 2011 O104:H4 outbreak strain. Besides avoiding antibiotics that can induce Stx production by triggering entry into the lytic cycle and therefore proliferation of *stx*-carrying lambdoid phages (Zhang *et al*, 2000; Bielaszewska *et al*, 2012), natural or synthetic compounds that can reduce c-di-GMP accumulation or curli synthesis or amyloid formation—that are currently sought after intensively (Cegelski *et al*, 2009; Sambanthamoorthy *et al*, 2012)—may show some benefit. Also probiotics may be an option, since the probiotic *E. coli* strain Nissle 1917 in co-culture with the O104:H4 outbreak strain was recently shown to reduce adherence to two human colonic epithelial cell lines and to attenuate Stx production and cytotoxicity of the O104:H4 outbreak strain (Rund *et al*, 2012; Wieler *et al*, 2014).

Finally, it may also be worth noticing that *E. coli* and *Salmonella* use biofilm matrix components including amyloid curli fibres and PGA to attach to plant surfaces (Barak *et al*, 2005; Jeter & Matthysse, 2005; Matthysse *et al*, 2008; Fink *et al*, 2012). High expression of these components in the O104:H4 outbreak strain is thus consistent with contaminated sprouts being the source of the strain during the 2011 outbreak (Buchholz *et al*, 2011) and the absence of the strain in a survey of cattle in the outbreak area (Wieler *et al*, 2011).

# Materials and Methods

### Bacterial strains

In this study, two clonal isolates of the *E. coli* O104:H4 outbreak strain were analysed, that is LB226692 (genome sequenced by Mellmann *et al* (2011)) and RKI II-2027 (designated by the Robert-Koch-Institut (Berlin) as the official outbreak strain). In addition, we used EAEC O104:H4 strain 55989 (Mossoro *et al*, 2002), HUSEC041 (or 01-09591, an enterohaemorrhagic O104:H4 strain isolated in 2001) (Mellmann *et al*, 2008), EDL933 (a classical EHEC O157:H7) (Riley *et al*, 1983; Perna *et al*, 2001) and the non-pathogenic *E. coli* K-12 strains W3110 (Hayashi *et al*, 2006) and AR3110, a direct derivative of W3110, in which cellulose synthesis was restored by replacing a stop codon in *bcsQ* by a sense codon (Serra *et al*, 2013a).

The non-polar deletion mutations in *bcsE*, *bcsF* and *bcsG* were generated by one-step inactivation with *kan* or *cat* resistance cassettes that were subsequently flipped out (Datsenko & Wanner, 2000). The oligonucleotides used in this procedure are listed in Supplementary Table S3.

The single-copy *dgcX::lacZ* reporter fusion inserted at the *attB* location of the chromosome of strain W3110 was constructed using chromosomal DNA of the outbreak strain LB226692 as a template for PCR with the primers listed in Supplementary Table S3. The procedure used was exactly as described for similar *lacZ* fusions to all GGDEF/EAL domain-encoding genes of *E. coli* (Sommerfeldt *et al*, 2009) that were used here for comparison of expression levels.

### Genome comparison

Genome sequences were obtained from the National Center for Biotechnology Information (NCBI) database. In order to find and identify novel GGDEF and EAL domain proteins, the Basic Local Alignment Search Tool (BLAST) (Altschul *et al*, 1997) was used to search for proteins encoded within the genomes of *E. coli* K-12 strain W3110, *E. coli* O104:H4 strain 55989, *E. coli* O104:H4 strain 01-09591 (HUSEC041), *E. coli* O157:H7 strain EDL933 and *E. coli* O104:H4 strain LB226692 (the 'outbreak' strain). When identifying possible GGDEF domain-containing proteins, the 169 amino acid long GGDEF domain of the YdaM protein (as identified in the Simple Modular Architecture Research Tool (SMART) database (Schultz *et al*, 1998)) was used as the query sequence. For the EAL domain search, the 245 amino acid long EAL domain of the YjcC protein (again defined by SMART) was used as the query sequence. Follow-up BLAST searches were performed to identify proteins with GGDEF and EAL domains too divergent from the YdaM and YjcC queries. Established GGDEF and EAL domain-containing proteins, found in the *E. coli* K-12 strain W3110, were used as query sequence in subsequent BLAST searches. If there were discrepancies in the amino acid sequence for a given protein between the various strains of *E. coli*, a genome comparison was performed, where the corresponding nucleotide sequence was multiply aligned using the CLUSTAL X programme (Larkin *et al*, 2007). The nucleotide sequences were examined for sequence variations (in particular deletions and insertions that may disrupt genes and/or lead to frameshifts or premature termination of translation) and incorrect initiation codon assignments with putative ribosomal binding sites also being taken into account.

The Geneious DNA sequence analysis software (version 5.4, Geneious) was used to generate multiple alignments and genome figures that were further processed using the GIMP2 programme (available at www.gimp.org). Relevant nucleotide variations in the genomes of the outbreak strain (and in relevant cases also of

EDL933) in comparison with the other strains were verified by custom sequencing (GATC GmbH, Konstanz) of PCR fragments generated with chromosomal DNA as a template.

## Growth of bacterial macrocolonies and visual detection of curli and cellulose formation

For the determination of macrocolony morphology, freshly prepared agar plates containing standard LB medium (Miller, 1972) or salt-free LB medium (LBnoS) were inoculated with 5 µl overnight culture applied as small spots. Plates were sealed with parafilm and incubated up to 14 days at the temperature indicated. For the detection of curli formation, LBnoS agar plates containing 40 µg ml$^{-1}$ of the amyloid-binding dye Congo red (Merck) and 20 µg ml$^{-1}$ Coomassie Brilliant Blue G (Sigma) were inoculated with 5 µl overnight culture and incubated for 7 days. Cellulose production was visualized by Calcofluor fluorescence of 3 days old colonies growing on solid media containing 0.01% Calcofluor Fluorescent Brightener 28 (Sigma). For photography of macrocolonies, a Leica S8 AP0 stereomicroscope and a Leica DC 300F digital camera were used.

## Protein purification

For the purification of C-terminally His6-tagged CsgD protein, the *csgD* gene was cloned into the expression vector pQE60 (Qiagen) using the oligonucleotide primers listed in Supplementary Table S3. The lacI$^q$ strain FI1202 (Fiedler & Weiss, 1995) containing pQE60:: *csgD* was grown in LB medium (100 µg ml$^{-1}$ ampicillin) at 37°C to an optical density (578 nm) of 0.5. CsgD protein expression was induced by adding IPTG (final concentration 1 mM), and cultivation was continued at 28°C for 4 h. Cells were resuspended in TE buffer (10 mM Tris–HCl pH 8.0, 1 mM EDTA pH 8.0), supplemented with 300 µg ml$^{-1}$ lysozyme and lysed using a French Press. CsgD accumulated in inclusion bodies was obtained by centrifugation and resuspended in 8 M urea. Purification by Ni-NTA affinity chromatography under denaturing conditions was carried out according to standard protocols (Qiagen). Eluted protein was dialysed against 300 mM NaCl and 50 mM NaH$_2$PO$_4$ pH 7.8 overnight at 4°C and used for SDS–PAGE.

## Immunoblots

Determination of cellular levels of CsgD in macrocolonies was performed by immunoblotting. Either entire macrocolonies were scraped off the agar plates or samples were taken from the indicated regions of the macrocolonies and resuspended in minimal medium M9 (Miller, 1972), and the optical density (578 nm) was determined. Sample preparation for SDS–PAGE and immunoblot analysis was performed as described previously (Lange & Hengge-Aronis, 1994). 7.5 µg cellular protein was applied per lane. A polyclonal serum against CsgD (custom-made by Pineda-Antikörper-Service, Berlin), goat anti-rabbit IgG alkaline phosphatase conjugate (Sigma) and a chromogenic substrate (BCIP/NBT; Boehringer Mannheim) were used for detecting CsgD protein. Quantification of CsgD levels was performed using ImageGauge V3.45 and purified CsgD-6His as a standard on the SDS gels.

### The paper explained

#### Problem

The 2011 German outbreak with a Shiga toxin (Stx)-producing *Escherichia coli* H104:H4 was not only the largest of its kind, but also that with the highest incidence (> 22%) of progression to haemolytic uraemic syndrome (HUS) ever recorded. Rapid genome sequencing showed the outbreak strain to be an enteroaggregative *E. coli* (EAEC) that had acquired a prophage carrying the Stx gene usually found in classical enterohaemorrhagic *E. coli* (EHEC). This led to speculations that its high virulence arises from a combination of the strong adherence to intestinal epithelial cells typical for EAEC with high Stx production. However, the virulence plasmid carrying the genes for EAEC-specific aggregative adherence fimbriae was later on found to be dispensable for colonization and causing bloody diarrhoea, leaving the question open again why the outbreak strain is so highly virulent.

#### Results

Based on detailed genomic comparisons—taking into account even single nucleotide polymorphisms with consequences for gene expression and regulation—complemented by experimental analyses, we report a unique combination of biofilm-associated properties in the O104:H4 outbreak strain not observed in any pathogenic *E. coli* before. These include: (i) the presence of two extra diguanylate cyclases producing the biofilm-promoting second messenger c-di-GMP, one of which (DgcX) is by far the most strongly expressed diguanylate cyclase observed to date in *E. coli*; (ii) high and deregulated production of the c-di-GMP-controlled biofilm regulator CsgD and amyloid curli fibres, a biofilm matrix component that promotes adherence to epithelial cells, at 37°C; (iii) frequent occurrence of derivatives with even further enhanced CsgD and curli production; and (iv) an inability to produce the exopolysaccharide cellulose.

#### Impact

These results may provide a key piece in the puzzle of understanding the high virulence of the 2011 outbreak *E. coli* H104:H4. Besides contributing to adhesion and colonization, these unique properties may also boost inflammation, since curli fibres are highly pro-inflammatory, while cellulose, which tightly associates with curli fibres in a composite material, counteracts this effect. Enhanced inflammation induced by 'naked', that is cellulose-free, curli fibres as produced by the outbreak strain could aggravate tissue damage and thereby promote more efficient transition of Stx into the blood stream resulting in an increased probability to develop HUS.

## Determination of β-galactosidase activity

β-galactosidase activity was assayed by the use of *o*-nitrophenyl-β-D-galactopyranoside (ONPG) as a substrate and is reported as µmol of *o*-nitrophenol per min per mg of cellular protein (Miller, 1972). Experiments showing the expression of *lacZ* fusions along the entire growth cycle were started by diluting an overnight culture into fresh medium to an optical density (OD$_{578}$) of 0.03, with sampling starting 2 h after inoculation. Such experiments were done at least twice, with a representative experiment being shown. Single-value data are the average of at least three measurements from independent cultures.

## Determination of spontaneous mutation frequency

Mutation rates were determined on the basis of spontaneous mutation to rifampicin resistance. Three 0.1-ml samples of each overnight culture were spread onto LB plates containing rifampicin (50 µg ml$^{-1}$). In parallel, the same cultures were used to determine

total viable cells in the absence of the antibiotic. Plates were incubated overnight at 37°C; colony-forming units were counted and averaged for the three samples from each culture. All experiments were done in triplicate with independent cultures.

**Supplementary information** for this article is available online: http://embomolmed.embopress.org

## Acknowledgements

We are grateful to H. Karch (Universität Münster) and Angelika Fruth (Robert-Koch-Institut, Wernigerode) for providing bacterial strains. We thank Alexandra Possling (Hengge group) for generating the mutants and the image shown in Supplementary Fig S6B and Christa Ewers and Inga Eichhorn (Wieler group) for practical support. Financial support was provided by the European Research Council under the European Union's Seventh Framework Programme (ERC-AdG 249780 to RH) and the Deutsche Forschungsgemeinschaft (GRK 1673).

## Author contributions

Concept of the study was made by RH. Design of the experiments was performed by AMR, TLP and RH. Experiments were performed by AMR and TLP. Bioinformatic analysis was carried out by TLP. Interpretation of data was made by AMR, TLP, LHW and RH. The paper was written by RH with input from all other authors.

## Conflict of interest

The author declares that they have no conflict of interest.

## For more information

Web page of the research group of Regine Hengge: http://mikrobiologie.hu-berlin.de/hengge.

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
