## [Review Process File · EMBO Molecular Medicine]

Cyclic-di-GMP signaling and biofilm-related properties of the Shiga toxin-producing 2011 German outbreak *Escherichia coli* O104:H4

Anja M. Richter, Tatyana L. Povolotsky, Lothar H. Wieler and Regine Hengge

Corresponding author: Regine Hengge, Humboldt Universität zu Berlin

Review timeline:

Submission date:	01 June 2014
Editorial Decision:	01 July 2014
Revision received:	12 September 2014
Editorial Decision:	23 September 2014
Revision received:	02 October 2014
Accepted:	06 October 2014

Transaction Report:

Editor: Céline Carret

1st Editorial Decision

01 July 2014

Thank you for the submission of your manuscript to EMBO Molecular Medicine. We have now heard back from the three referees whom we asked to evaluate your manuscript. Although the referees find the study to clearly be of interest, they also raise concerns that must be addressed in a revised version of your article.

As you will see from the comments below, although referee 3 is supportive, referees 1 and 2 are much more critical. They both suggest adding some experimental data and I would strongly recommend you to test your hypothesis experimentally as mentioned by referees 1 and 2. In addition, referee 1 would appreciate an extended comparative genomics analysis including more EHEC and EPEC strains. Indeed, no EPEC strain was included and we agree that this would strengthen this part of the study.

Given the balance of these evaluations, we feel that we can consider a revision of your manuscript if you can address the issues that have been raised. Please note that it is EMBO Molecular Medicine policy to allow only a single round of revision and that, as acceptance or rejection of the manuscript will depend on another round of review, your responses should be as complete as possible.

EMBO Molecular Medicine has a "scooping protection" policy, whereby similar findings that are published by others during review or revision are not a criterion for rejection. Should you decide to submit a revised version, I do ask that you get in touch after three months if you have not completed

it, to update us on the status.

I look forward to seeing a revised form of your manuscript as soon as possible.

***** Reviewer's comments *****

Referee #1 (Remarks):

The paper of Richter et al. reports the characterisation of cyclic-di-GMP signalling and biofilm-related properties in the 2011 German outbreak *Escherichia coli* O104:H4 strain. To understand why this strain was so virulent is clearly a topic of interest. The work is well done, however to my opinion, we still have an incomplete story.

Here are my comments.

The part of comparative genomics needs to be performed with more strains. Due to the huge diversity of *E. coli*, comparative genomics on complete genomes should be extended to B1 phylogroup strains of EHEC and EPEC pathotypes as well as to B1 commensals. Numerous genomes are available.

Page 11, second paragraph on the mutants with the CsgD and curli content mutants. The authors should sequence some strains to identify the mutations involved.

Page 15, second paragraph, In conclusion... The hypothesis that the observed properties can boost adhesion and colonisation should be tested.

Minor points.

Page 13, second paragraph, the German strain is of B1 phylogroup (not A).

Page 19, mutation frequency. Experiment should be repeated at least three times independently, and the median used.

Referee #2 (Remarks):

These results correlate specific genetic properties of *E. coli* O104:H4 with its unusual virulence. The main traits analyzed are the production of biofilm and curli fibers, both positively regulated by C-di-GMP whose levels seem to be increased due to elevated expression of the *dgcX* gene. Expression of this gene is dependent on RpoD or RpoS, ensuring high amounts of the signaling molecule both during exponential and stationary growth. Additionally, the pathogen does not produce cellulose, a mechanism that counterbalances the effect of curli fibers on adhesion and inflammation. Together, these traits may be among the reasons why O104:H4 caused a severe and deadly outbreak.

Overall, the manuscript reports excellent science and it is well written. However, a few points can be criticized, the main one being that most of the correlations between the genetic work described in the manuscript and the actual course of the disease rely on observations reported in other publications. Although generally valid, the authors may consider including some direct evidence showing differential effects on the infection caused by the original strain compared to derivatives in which one or more of the genes involved in the phenotype, as *dgcX* or *rpoS*, were impaired.

The lack of direct results is of particular concern when a potential treatment for the infection is proposed. In my view, the observed correlations, in particular the potential use of a probiotic strain

should merit some direct experimental evidence.

The procedure to grow the bacterial cultures should be described and proof that they were in exponential phase at the beginning of the experiment (relevant in a figure referred as E2 in p7) should be included.

A minor point is that the reader would be greatly helped to follow the long and at times complex discussion if an illustrative diagram would be provided. Equally, numbering of the figures would help to read the manuscript.

Referee #3 (Remarks):

This paper sheds light on an interesting physiological aspect of strain *Escherichia coli* (EHEC) of the serotype O104:H4 that was responsible for the major outbreak of bloody diarrhea in 2011. While most of the studies on this strain focused around the shiga toxin, this study shows the interesting role of curli fibers in this disease. Curli fibers are important not only in adhesion, but also in internalization of the bacteria by epithelial cells and therefore the fact that they are overexpressed by the epidemic strain explains much of the virulence of these bacteria. Personally, I am quite intrigued by the fact that the strain constantly improves Curli production by successive mutations. This is clearly an interesting direction for future studies.

1st Revision - authors' response

12 September 2014

Referee #1 (Remarks):

The paper of Richter et al. reports the characterisation of cyclic-di-GMP signalling and biofilm-related properties in the 2011 German outbreak *Escherichia coli* O104:H4 strain. To understand why this strain was so virulent is clearly a topic of interest. The work is well done, however to my opinion, we still have an incomplete story.

Here are my comments.

The part of comparative genomics needs to be performed with more strains. Due to the huge diversity of *E. coli*, comparative genomics on complete genomes should be extended to B1 phylogroup strains of EHEC and EPEC pathotypes as well as to B1 commensals. Numerous genomes are available.

A: For the newly discovered diguanylate cyclase DgcX, which we initially detected in the outbreak strain and the two closely related EAEC strains 55989 and HUSEC041, we have extended our analysis to include 61 available E. coli genomes sequences (from all pathotypes and phylogroups). These data are now shown in a new Fig. 1 and described in detail on p. 6ff. In short, we found the dgcX gene at the same chromosomal location also in a strain related to the 2011 outbreak strain that was isolated in Georgia already in 2009 as well as in two ETEC (which even belong to different phylogroups). In addition, there is a commensal E. coli that carries it too, but at a different chromosomal site. In all cases, dgcX is inserted at the right end of various lambdaoid prophages strongly suggesting its horizontal transfer by 'hitchhiking' on these phages. The inclusion of these new data has now led to the genomic data being presented in two figures (Figs. 1 and 2).

Furthermore, we are in the process of searching and analyzing all genes for c-di-GMP-related proteins/enzymes in all of these 61 available genomes. This analysis is currently providing a series of interesting novel diguanylate cyclases and c-di-GMP-specific phosphodiesterases as well as small but consequential variations in the sequences of known such enzymes in different pathotypes and phylogroups of E. coli. These data, however, are beyond the scope of the study under review

here, which focuses on the 2011 outbreak strain, and will be published as a separate (again very detailed) study.

Page 11, second paragraph on the mutants with the CsgD and curli content mutants. The authors should sequence some strains to identify the mutations involved.

A: We obtained raw sequencing data of chromosomal DNA of the eight purified strains shown in (what is now) Fig. 8B, when one of us (L.W.) was on a recent sabbatical at the Sanger Institute (UK). Unfortunately, these sequence data were not of the high quality required to unequivocally identify single mutations, but contained quite a high number of variations in comparison to the sequence of the outbreak strain isolate LB226692. Even after discarding (i) all cases of +/- 1 bp in stretches of a specific nucleotide as 'standard' sequencing errors or (ii) specific variations to LB226692 present identically in all 8 derivative strains (that point to occasional sequencing errors in the LB226692 genome sequence in the data base), there remained on average >30 positions/Mbp with sequence variations, which made it impossible to pinpoint real and specific mutations. On the background of this high rate of obvious sequencing errors, we unfortunately could also not detect variations that correlated with the three types of colony morphology that differ from that of the original strain (see text on p. 12 and Fig. 8B).

Overall, the results with the 'red sector derivatives' in our study show that (i) the underlying changes are stable, i.e. are due to mutations; (ii) these changes result in increased CsgD and curli expression and thus in altered colony morphology; (iii) the mutations are not located in the promoter region of csgD, i.e. they act in trans on CsgD expression (this finding is based on local high quality sequencing). Since the mutations lead to high-level expression of CsgD/curli at 37°C they overrun the temperature regulation of CsgD, the mechanistic basis of which is still a mystery (despite of our detailed knowledge of sigma factors, transcription factors and complex c-di-GMP-mediated control acting at the csgD promoter as well as of an entire series of sRNAs affecting csgD mRNA). Moreover, the high frequencies, at which these mutations come up in the outbreak strain as well as in EAEC 55989, points to a yet uncharacterized mechanism operating in distinct E. coli strains that provides for rapid phenotypic switching of expression levels of CsgD and therefore curli fibres at 37°C. We'll investigate this unknown mechanism in the future, but it is clear that this requires an entire new study.

Page 15, second paragraph, In conclusion... The hypothesis that the observed properties can boost adhesion and colonisation should be tested.

A: This is not a hypothesis. Rather, c-di-GMP generally controls the expression or biosynthesis of adhesins and/or adhesive/aggregative biofilm matrix components in most bacteria where c-di-GMP targets have been studied (summarized in reviews e.g. by Jenal & Malone, 2006; Hengge, 2009). Amyloid curli fibres in particular – which are aggregative and interact with fibronectin and laminin – have been shown to stimulate adhesion and colonisation directly (Gophna et al, 2001; Gophna et al, 2002; Olsén et al, 1989; Olsén et al, 1998; Rosser et al, 2008; Saldana et al, 2009).

Overall, this sentence is meant as a summarizing statement close to the end of the Discussion, with all these relevant references being cited along with the details on the previous page.

Minor points.

Page 13, second paragraph, the German strain is of B1 phylogroup (not A).

A: This was a typo that was corrected. We now also refer to phylogroups on p. 6, where we present the data on dgcX searched for in 61 genome sequences.

Page 19, mutation frequency. Experiment should be repeated at least three times independently, and the median used.

A: We have repeated the experiment again (which fully confirmed the previous data that had been based on two independent experiments only), so that Fig. 9 now includes data from three completely independent experiments. We have decided to show the direct results of the three experiments for each of the six strains tested rather than the average or the median (now Fig. 9).

Referee #2 (Remarks):

These results correlate specific genetic properties of E.coli O104:H4 with its unusual virulence. The main traits analyzed are the production of biofilm and curli fibers, both positively regulated by C-di-GMP whose levels seem to be increased due to elevated expression of the *dgcX* gene. Expression of this gene is dependent on RpoD or RpoS, ensuring high amounts of the signaling molecule both during exponential and stationary growth. Additionally, the pathogen does not produce cellulose, a mechanism that counterbalances the effect of curli fibers on adhesion and inflammation. Together, these traits may be among the reasons why O104:H4 caused a severe and deadly outbreak.

Overall, the manuscript reports excellent science and it is well written. However, a few points can be criticized, the main one being that most of the correlations between the genetic work described in the manuscript and the actual course of the disease rely on observations reported in other publications. Although generally valid, the authors may consider including some direct evidence showing differential effects on the infection caused by the original strain compared to derivatives in which one or more of the genes involved in the phenotype, as *dgcX* or *rpoS*, were impaired.

A: Ideally, the 'some direct evidence' suggested to be included should be experiments in which an animal model (germ-free mice and infant rabbits have so far been used in two studies, i.e. by Al Safadi et al. 2012, and Munera et al. 2014, resp.) is infected with the outbreak strain as well as specific knockout mutants generated from this strain.

Please, note however, that we present here a microbiological study describing c-di-GMP signaling and biofilm-related properties of the 2011 outbreak E. coli O104:H4 strain, with data being presented in 9 Figures and 6 supplementary Figures. It is not before the Discussion, that we relate part of these data (mainly very high curli expression in the absence of cellulose at 37°C) to previously published knowledge on the strong inflammatory effect of curli fibres that is counteracted by the presence of cellulose, pointing out the implications with respect to the high virulence of the 2011 outbreak strain. We can fully understand that the reviewer was so intrigued by these implications, that he/she wishes to see immediate experimental confirmation. We were equally intrigued, when we realized this impact of our data and we agree that it would be wonderful to have direct confirmation from animal experiments.

However, please, take into account that:

- *In Germany the O104:H4 strain has been classified as a security level 3 strain – while we can handle the strain in microbiological assays in the lab of one of us (L.W.), we do not have an animal facility at our disposal that would allow us to do animal experiments with S3 level bacteria.*
- *Even if we had the technical opportunity to do this kind of animal experiments (which in the future may actually be the case at some point for the Wieler group), we believe that doing such experiments properly requires a full novel study (with many appropriate controls and also including probiotic strains that may have protective effects) that is beyond the scope of our microbiological study presented here.*

In conclusion, we currently do not have the technical possibility to do such experiments. However, we actually hope that some colleagues with access to the necessary facilities – who will see our

results and their implications – will be equally intrigued and try to do this kind of animal experiments.

The lack of direct results is of particular concern when a potential treatment for the infection is proposed. In my view, the observed correlations, in particular the potential use of a probiotic strain should merit some direct experimental evidence.

A: Again, this would require the animal experiments described above (co-infection with the outbreak strain and the probiotic EcN) that we do not have the possibility to do. However, co-culture effects of the two strains in vitro and on cell cultures have actually already been reported (see Rund et al. 2012, and – just accepted – Wieler et al. 2014). These papers are both mentioned in the Discussion, with any unnecessary speculation having been removed.

The procedure to grow the bacterial cultures should be described and proof that they were in exponential phase at the beginning of the experiment (relevant in a figure referred as E2 in p7) should be included.

A: We routinely start our cultures at an initial OD(578 nm) of 0.02 – 0.03 and grow them for at least two hours (>3 generations in LB) before we begin sampling for any kind of assays. This is now described in more detail in M&M.

A minor point is that the reader would be greatly helped to follow the long and at times complex discussion if an illustrative diagram would be provided. Equally, numbering of the figures would help to read the manuscript.

A: We have shortened and streamlined the Discussion to now focus on the essential points only. We hope that this improves readability. Since this study is not about e.g. a specific molecular mechanism, we do not see how to summarize it in a single diagram. We are sorry for the numbers of the figures apparently not showing up in the pdf files available to the reviewers. In fact, also a wrong figure number had been given in the text a couple of times, which may have added to confusion. This was corrected.

Referee #3 (Remarks):

This paper sheds light on an interesting physiological aspect of strain Escherichia coli (EHEC) of the serotype O104:H4 that was responsible for the major outbreak of bloody diarrhea in 2011. While most of the studies on this strain focused around the shiga toxin, this study shows the interesting role of curli fibers in this disease. Curli fibers are important not only in adhesion, but also in internalization of the bacteria by epithelial cells and therefore the fact that they are overexpressed by the epidemic strain explains much of the virulence of these bacteria. Personally, I am quite intrigued by the fact that the strain constantly improves Curli production by successive mutations. This is clearly an interesting direction for future studies.

A: Yes, we think so, too, and will proceed to do such studies in the future (see also comments above).

Overall, we thank all the reviewers for their constructive comments.

2nd Editorial Decision

23 September 2014

Thank you for the submission of your revised manuscript to EMBO Molecular Medicine. We have now received the enclosed reports from the referees who were asked to re-assess it. As you will see the reviewers are now supportive and I am pleased to inform you that we will be able to accept your manuscript pending the following final amendments:

Please, could you follow referee #2 recommendations and provide an improved discussion? The addition of a schema is not required, but the discussion should be clear even for non-specialists.

Please submit your revised manuscript within two weeks. I look forward to seeing a revised form of your manuscript.

***** Reviewer's comments *****

Referee #1 (Remarks):

The paper has been improved.

Referee #2 (Remarks):

I still find that there is plenty of room for improvement to clarify the discussion. I am sure that the authors can make an extra effort.

Regarding the reply to my first comments I disagree with the author. First, an increase in the amount of data cannot be claimed as providing validation for their indirect nature. Second, although I fully sympathize with the author, the lack of facilities or equipment at the authors laboratory is not a scientific reason to justify the lack of pertinent data.

A minor point is that numbering of figures, after the modifications introduced in the revision, contains errors (applies to Fig 1 and 2).

2nd Revision - authors' response

02 October 2014

In response to your letter from September 23, 2014, we have introduced the following final changes:

"Please, could you follow referee #2 recommendations and provide an improved discussion? The addition of a schema is not required, but the discussion should be clear even for non-specialists."

The Discussion has now been further shortened. I took out all special details (these can be found in the Results for those who are want to get down to such detail), tried to use a more general language and now focus on the major conclusions and implications on the background of previous knowledge on the outbreak strain and other relevant pathogenic E. coli. It is probably the shortest and most concise discussion I have ever written, but indeed it now reads much better than in the longer version.

In response to the final comments of reviewer 2, the following was changed:

"I still find that there is plenty of room for improvement to clarify the discussion. I am sure that the authors can make an extra effort."

See above.

"A minor point is that numbering of figures, after the modifications introduced in the revision, contains errors (applies to Fig 1 and 2)."

Errors were corrected (and see above).

I am confident that these final changes have further improved the manuscript. Thank you for your patience and the handling of our manuscript.